# Risk and Protective Factors in Ecuadorian Adolescent Survivors of Suicide

**DOI:** 10.3390/children10030549

**Published:** 2023-03-14

**Authors:** Marly Johana Bahamón, José Julián Javela, Stefano Vinaccia, Shadye Matar-Khalil, Andrés Cabezas-Corcione, Evelyn Esther Cuesta

**Affiliations:** 1Psychology Program, I-Flor Research Group, Faculty of Health Sciences, University of Sinú, Montería 230002, Colombia; 2Psychology Program, Faculty of Health Sciences, Technological University Indoamérica, Ambato 180103, Ecuador

**Keywords:** suicide survivors, suicide, suicidal behavior, self-injury, adolescent

## Abstract

Suicide is one of the main causes of death among the adolescent population, which is why it is considered an important mental-health problem. In addition to this situation, for each suicide, the group of people who survive it (known as suicide survivors) can present serious emotional affectations, becoming a population at risk for this problem. The purpose of this study was to analyze the effect of suicide-survivor status on risk factors and protective factors for suicide. A total of 440 adolescents with a mean age of 15.78 (SD = 1.74) participated, who were divided according to survivor status, identified as the SV group (79 cases), and non-survivors, identified as the NSV group—adolescents that did not have experience or contact with a suicide attempt (361 cases). A questionnaire of sociodemographic characterization and risk conditions, the Alexian Brother Urge to Self-Injure (ABUSI), the Plutchik Suicide Risk Scale, the Multidimensional Scale of Perceived Social Support, and the Cognitive and Affective Empathy Test (TECA) were applied. Descriptive statistics, mean difference for independent samples, contingency tables, X2 statistic, Fisher’s exact statistic, and Cohen’s d coefficient were used. The results show significant differences between SV and NSV participants in risk and protective factors regarding the presence of a greater adoption of perspective and emotional understanding. On the other hand, NSV adolescents presented higher scores of perceived social supports regarding risk factors, and there was a higher proportion of a history of suicide attempt, severity/hospitalization, impulse to self-harm, and level of suicidal risk in the SV group. The need to incorporate forms of suicide prevention with the survivor population is discussed, increasing the possibilities of postvention.

## 1. Introduction

Suicide is the second leading cause of death in subjects between 15 and 29 years of age and is considered one of the main mental-health problems during adolescence, with rates that have remained relatively stable in recent decades in most of the world’s population [1,2,3]. In Ecuador, adolescent suicide rates are between 7.5 and 13.6 per 100,000 inhabitants, representing between 15% and 20% of deaths from external causes in this age group [4]. This problem is considered a multifactorial phenomenon in which different factors converge, among which previous suicide attempts have been identified as important clinical predictors [5], as well as mood disorders [1], neurobiological factors [6], the use of psychoactive substances, family history of suicidal behavior [7], impulsive aggressiveness, negative emotions [8], adverse life experiences [9,10] and self-injurious behaviors [11].

The suicide phenomenon is so complex and common that many humans will have some experience related to this issue throughout their lives. In the United States it has been identified that 51% of adults are exposed to at least one suicide during their lifetime, of which 35% have reported severe or moderate grief from the loss [12].

Thousands of families lose a loved one to suicide each year, having to deal with, in addition to the loss, the stigma, which can further burden them and impede the grieving process. These individuals are considered suicide survivors; however, today, in addition to family members, a suicide survivor is a person who has lost a significant family member or friend to suicide [12], or any person who is exposed to or who has been significantly affected by the suicide of a close one [13,14]. 

Studies have reported that suicide survivors have more risk factors for their emotional health, and are more vulnerable to emotional distress and psychopathology such as depression, post-traumatic stress disorder, complicated grief, and even suicidal ideation and behavior [15,16]. In research where psychological autopsies and interviews with survivors have been conducted, mental-health problems have been identified before and after the death of the individual in more than half of their families, as well as many cases of attempted and completed suicide among survivors, which may be associated with genetic factors, stressors, and shared environments, with the conclusion that survivors are at high risk for suicide [17,18]. 

Suicide survivors tend to experience different negative emotions linked to abandonment, rejection, and guilt out of fear that the suicide death was a retaliation by the deceased [19]. In addition to the pain of the death of a loved one, they must deal with stigma from their community and social networks, which are based on prejudices rooted in the idea that survivors are guilty, contaminated, or mentally disturbed [20,21]. 

Much of the research with this population has focused on suicide bereavement and its relationship to clinically significant psychological distress [14,21,22]. This is linked to the social perception that the act of suicide is a failure of the victim and the family to cope with emotional problems. In this regard, the findings suggest that close survivors of the deceased have a higher risk of suffering a complicated bereavement [23], although little is known about whether close kinship also represents an increased risk of complicated bereavement [24,25].

In relation to protective factors in survivors, the findings are scarce, and recent contributions have focused on social support, understood as the set of material, informational, and psychological resources provided by the subject’s immediate social network to make facing stressful situations easier. This includes tangible emotional support, spaces to share common interests, and conditions for the subject to feel understood and respected, which seems to work as a protective factor for coping with grief when finding that relationships that significantly reduce feelings of loneliness and sadness [26]. Studies with suicide survivors in this population have offered relevant information on the possibilities of intervention in coping, which have focused on problematic grief, considering that it is one of the most common complications among its victims and generates the greatest psychological discomfort. Problematic grief is one of the most analyzed variables in suicide survivors. However, recent evidence suggests that it develops independently of the symptoms of depression and post-traumatic stress disorder [27]. 

Research on suicide survivors is not prolific and most of the studies have been carried out with an adult population, leaving possibilities to explore the psychological response of other age groups at higher risk, such as adolescents. Complex transition processes arise at this stage, and few studies have focused on the experience of adolescent survivors and the impact this may have on their coping and future adaptation, since in addition to facing the vital crisis of adolescence they must assume the adverse event of suicide [24,27].

Considering the multiple possibilities offered by investigating the characteristics of survivors in a critical stage such as adolescence and the existing empirical vacuum, the aim of this research was to analyze the effect of being a suicide survivor in Ecuadorian adolescents between 12 and 18 years of age on risk factors and protective factors.

The hypotheses raised were the following: (a) The drive for self-injurious behavior and suicidal risk is greater in adolescent suicide survivors than in non-survivors, (b) there are significant differences in cognitive and affective empathy between surviving and non-surviving adolescents, and (c) perceived social support is higher in non-surviving adolescents compared to survivors.

## 2. Method

### 2.1. Participants

There was an incidental sample of 440 adolescents between 12 and 18 years of age with a mean of 15.78 (SD = 1.74), with place of residence in the province of Tungurahua, a region with one of the highest suicide rates in Ecuador. The sample was divided according to the condition of survivor (people who have lost a family member or close person by suicide) identified as the SV group, and a second group not exposed to the condition of survivor identified as the NSV group. The SV group consisted of 79 cases (M = 16.16, SD = 1.54), of which 39 (49.4%) were men and 40 (50.6%) were women, and the NSV group comprised 361 cases (M = 15.70, SD = 1.77), of which 184 were men (51%) and 177 (49%) women. 

The inclusion criteria were the age range being met, legal permission from the parents or guardians, the voluntary nature of the participants, and a level of schooling in accordance with the age that would allow the understanding of the questions contained in the instruments. Adolescents who did not have a basic level of academic training to understand the questionnaires were excluded.

### 2.2. Instruments

*Characterization questionnaire*: set of questions referring to age, sex, area of residence, economic level, status as a survivor from having a family member or close person losing his or her life by suicide, and previous suicide attempt and severity of the attempt by determining whether he or she had been hospitalized for this reason.

*The Alexian Brother Urge to Self-Injure (ABUSI)* [28]: an instrument that evaluates the cognitive and emotional aspects of the intensity of the impulse to self-injure in the previous week by assessing its frequency, intensity, and duration. Its structure is unidimensional and it is composed of 5 items with 7 response options scoring between 0 and 6. The internal consistency reported by the authors ranged between α = 0.92 in the initial measurement and α = 0.96 in the second measurement. All its items were highly correlated with the overall scale (0.87 to 0.92). The cutoff point to consider a score high is 6 or more.

*The Plutchik Suicide Risk Scale:* validated in Colombia by Suárez-Colorado et al. (2019) and assesses previous attempts at self-harm, intensity of current suicidal ideation, feelings of depression, hopelessness, and others [29]. It has 15 items and is scored by giving a value of 1 to all affirmative responses and 0 to negative responses. Scores above 6 indicate suicide risk. Two factors were found in the Colombian adolescent population: suicidal risk (items 13, 14, 15) and depressive symptoms (items 2, 3, 6, 8, 9, 10). Cronbach’s-alpha reliability for depression was 0.72 and 0.80 for suicide risk; McDonald’s omega was 0.82 and 0.94, respectively. The AFC showed good fit (x2S-B = 26.36, gl = 26, *p* = 0.34; NNFI = 1.0, CFI = 1.0, RMSEA = 0.02, 90% CI (0.00, 0.05)).

*The Multidimensional Scale of Perceived Social Support* (validated by Trejos et al. 2018): consists of 12 items that evaluate the perception of social support through three dimensions: family (items 3, 4, 8, 11), friends (items 6, 7, 9, 12), and important people (items 1, 2, 5, 10). These 12 items have a 7-point Likert-type scale ranging from strongly disagree (1) to strongly agree (7). Its internal consistency is 0.84 (95% CI = 0.83–0.86), and the AFC indexes show good fit (AGFI = 31,680.98, BIC = 31,824.74, NNFI = 0.946, CFI = 0.975, RMSEA = 0.049) [30].

*Cognitive and Affective Empathy Test (TECA) (López-Fernández & Fernández-Pinto, 2008)*: consists of 33 questions that assess empathic happiness, perspective adoption, empathic stress, and empathic happiness. It has a 5-option Likert scale (1 “Strongly disagree” to 5 “Strongly agree”). The Cronbach’s alpha reported by the authors was 0.86 with the items and an explained variance of 37.4% [31].

### 2.3. Procedure

The group of researchers presented the research project to the ethics committee of a university in Colombia to request its approval. This committee granted approval on 27 August 2019 with code CEI-USB-CE-0294-00-00. Through the university welfare service, 15 research assistants were linked: students in the last semester of the psychology degree who personally contacted each of the participants. All suicide cases from the last five years in a city in Ecuador reported as having the third highest number of suicides in the country were investigated. Affected individuals were contacted by telephone and only adolescents whose legal guardians were accessed participated. They were informed about the objective of the investigation in order to obtain the informed consent of the parents or legal guardians, after which a personal and individual appointment was made for the application of the instruments, which were completed with pencil and paper in a neutral place. For future contact with the participants, they were given the data of the principal investigators and university welfare.

### 2.4. Analysis of Results

Descriptive statistics (means, standard deviations, and percentages) were used to analyze the socio-demographic characteristics and risk factors of the participants. Contingency tables and the X^2^ statistic were used to identify differences in means for independent samples (SV vs. NSV group). The X^2^ test was applied taking into account that it was possible to perform the analyses with samples of unequal size since the analyses are developed from the expected frequencies, through which the frequency distribution of the total number of cases was obtained. For expected frequencies less than 5, Fisher’s exact statistic was used since such small frequencies can lead to a reduction in statistical power. Finally, the effect size was determined using Cohen’s d coefficient. The data were analyzed using IBM SPSS^®^ 25.0 (IBM Corporation, Armonk, NY, USA).

## 3. Results

The sociodemographic characteristics identified in the adolescents surveyed show that most of the participants were between 16 and 17 years of age and came from low and middle socioeconomic strata, with similar representation of males and females, and mostly from urban areas. No significant differences were identified in any of the socio-demographic characteristics explored (Table 1).

Regarding the risk factors analyzed, the results show significant differences for all the variables explored, indicating a higher percentage of adolescent survivors with a history of suicide attempt (24.1%), severity of suicide attempt/hospitalization due to attempt (17.7%), and family history of suicide attempt (43%). When analyzing the effect of being a suicide survivor on the variables evaluated, large effect sizes were found for the severity of the attempt/hospitalization (*f* = 0.708) and small effect sizes for the history of suicide attempt (*f* = 0.432) (Table 2).

The analysis of cognitive and affective empathy in adolescents showed significant differences for the following dimensions: perspective adoption (*p* = 0.000; 65.8%) and emotional understanding (*p* = 0.000; 68.4%), with higher scores in the SV group compared to the NSV group. Regarding the effect of survival condition on the dimensions of cognitive and affective empathy, no significant sizes were found (Table 3).

Regarding self-harm impulse and the level of suicidal risk present in adolescents, higher scores were identified in the SV group (*p* = 0.001; 31.6%) with respect to the NSV group (16.1%), and higher suicidal risk in the SV group (*p* = 0.001; 41.8%) compared to the NSV group (23.3%), with statistically significant differences and large effect sizes of survivor status on these variables (self-harm impulse *d* = 1.116; suicidal risk *d* = 1.200), indicating that survivors are more likely to experience self-harm impulses and suicidal risk (Table 4).

The data found on perceived social support showed higher scores on perceived social support in NSV compared to SV adolescents; however, the differences were not significant and no significant effect sizes of survivor status on perceived social support were found (Table 5).

## 4. Discussion

This research explored different risk and protective factors among adolescent suicide survivors in order to provide more information on relevant aspects that may substantially affect this population and to provide tools for addressing their mental health. 

In this sense, the results found show that there were no differences between the sociodemographic characteristics of the participants according to the condition of survivor, evidencing that this is a problem that affects the entire population without discriminating between age range, economic level, and sex, among other conditions. This shows that the population exposed to suicide risk is increasingly diverse, which is evidenced precisely by the high number of people who have been exposed to a suicide-related experience during their lifetime [3,32]. 

The data from this research identified a higher frequency of suicide attempts among survivors compared to non-survivors, which highlights the greater mental-health impairment of survivors and the greater risk for suicidal behavior among those who have been exposed to the suicidal experience. In fact, the effect size of survivor status was large for the suicide-attempt variable, which is consistent with data from several studies that noted the increased risk in this population [26,32,33,34,35,36,37,38].

Abbott and Zakriski (2014), reported that the risk of suicide after exposure to another suicide in adolescents is two to four times higher than in any other age group, which is related to the severity of the suicide attempt among adolescent survivors, which was much higher with respect to the group of non-survivors, indicating the need to evaluate and follow up on this population group in order to prevent future suicide attempts and their subsequent success [39,40]. Likewise, survivors showed a greater impulse to self-harm and a high level of suicide risk, indicating that survivors are more prone to risky behaviors, significantly increasing the chances of carrying out suicidal behaviors in the future. 

In this regard, several authors have argued that it is likely that a large proportion of suicide survivors require support, but do not seek help or have limitations in finding and/or accessing it, and experience difficulties that stigma about the issue can create [26,32]. In this sense, beliefs about suicide among the population can increase stigmatizing behaviors towards survivors, significantly limiting their chances of recovery from loss and causing greater feelings of guilt and loneliness. Thus, stigma becomes a risk factor since subjects may come to consider that suicide cannot be prevented by limiting loved ones from intervening in a timely manner [39].

Regarding possible protective factors among adolescent survivors, the data found that survivors tend to adopt a greater emotional perspective and understanding than non-survivors. These two elements (understanding and perspective taking) refer in general terms to empathy, that is, the feeling that a person can experience when putting themselves in another’s place. In this line, the results found show that the survivors presented higher scores in these two components, evidencing a greater ability to understand other people’s emotional states. In other words, the experience of loss by suicide seems to promote the empathic capacity of individuals, and in the relational context this ability can favor subsequent help relationships that are involved in scenarios such as support groups.

This finding highlights the need to explore multiple characteristics that promote mental health among suicide survivors, given that studies have generally focused on support groups but have not shown core aspects such as quality of life, subjective well-being, perception of the future, and coping that would point to possibilities of accompaniment and guidance in psychological care. Likewise, the few studies that have explored different variables in this population are focused on the family nucleus and do not contemplate other support networks that can be used to increase the chances of the optimal development of survivors, taking advantage of their higher levels of empathy [37].

Social support presented higher scores among NSV participants, indicating that the possibilities of access to material and psychological resources perceived by individuals may be a mediating factor between different emotional crises and suicidal behavior. In this sense, some authors [30] have warned that it is not only a matter of having social-support resources, but also of being positively perceived by the subject in such a way that they make use of these. In the case of suicidal behavior, one of the most associated risk factors is isolation, and the results of this study lead us to infer that the greater the perception of social support, the lower the chances of carrying out suicidal behavior. In the case of the SV group, this perception may decrease, given the exposure to stigma and signaling by the community, leaving them more vulnerable after the experience of death by suicide of a loved one.

In this line, other studies have shown that social support is related to protective factors against suicide, including prevention programs [12,18,24,35]. Despite the stigmatizing attitudes that suicide survivors must face, which makes them prone to isolation and, therefore, to new risk factors for suicidal behavior, health professionals cannot lose sight of the need to carry out prevention and intervention processes aimed at reducing isolation by taking advantage of the greater empathic capacity in these individuals. Other studies have shown that social support is related to protective factors against suicide, including prevention programs. Likewise, secure attachment styles help in coping with loss and complicated grief, such as talking about feelings and potential personal skills [37].

Determining in-depth exposure to suicide is an important factor to focus on in postvention, research, and clinical practice, since it allows the establishment of evaluation indicators to identify those most affected and prone to developing mental disorders and increased risk of suicidal behaviors later [41]. The results of this investigation highlight the need for follow-up with suicide survivors who can be referred to mental-health services in a timely manner to reduce their risk of death by suicide compared to those who are not referred [35].

A little-explored aspect that is evident is accompaniment by health professionals in suicide-prevention processes, not only for suicide survivors but also for adolescents in general who may have limited access to health services if they are at risk. In fact, it is necessary to consider that this age group is mobilized in physical and virtual settings; therefore, the presence of mental-health professionals is vital to provide relevant information and advice regarding risk factors for suicide. In addition, this study shows the need to identify and make visible people exposed to suicide in order to acquire grief support and reduce the risk of suicide in the general population, especially since the WHO has established that for every suicide there are around 10 people seriously emotionally affected by the death of a loved one [32,38,42,43,44,45].

Developing studies with survivors constitutes a challenge for mental-health professionals and researchers, since there are still false beliefs about suicide that prevent the disclosure of accurate information, as well as stigmatizing behaviors that make this population invisible due to fear of rejection and finger pointing. Added to the aforementioned, to access adolescents it is necessary to have the consent of the parents, which is another barrier because it is necessary for them to have overcome different obstacles to authorize their children to speak about the subject.

In addition to the above, in the time elapsed between the death of the loved one and the moment of evaluation some investigations have reported a greater presence of complicated grief among suicide survivors [12,15,16], showing that the response of collaborating with and participating in an investigation can be significantly reduced based on the experience of grief.

Regarding the limitations of this research, the results must be taken with caution since the selection of the subjects was not random and it is possible that some of them had homogeneous sociodemographic characteristics, limiting the analysis of segments. Another aspect that should remain in similar studies is focused on exploring the relationship with the loved one who committed suicide, since the closeness of the bond can show different effects.

Due to the difficulties of accessing this population due to them being survivors and also minors, a limitation of this research was the small number of participants, which did not allow other statistical analyses such as measurement models to determine the factorial invariance of the evaluated aspects. In the two groups (SV and NSV), an aspect that subsequent studies can include is whether they have access to a significant number of survivors, since to carry out this type of analysis large samples are required in order to avoid imprecise estimates in standard errors and fit indices.

Future research can focus on the analysis of protective and risk factors to strengthen the development of psychological care and intervention programs that promote mental health in this population, as well as aspects such as the type of relationship with the suicide victim, the time elapsed between the suicide and the evaluation, and the support that the adolescent has received. Finally, considering mental-health measures through longitudinal studies in this population would offer clues about the approach and the key moments of prevention and intervention. 

## 5. Conclusions

We found that there are significant differences between surviving and non-surviving participants in risk and protective factors. Regarding protective factors, it was found that the SV group presented higher perspective-adoption and emotional-understanding scores than the NSV group, indicating greater empathic capacity in adolescents who have experienced the loss of a loved one to suicide. Perceived social support yielded higher scores in the NSV group of adolescents, although the differences found were not significant as expected.

Regarding risk factors, there was a higher proportion of a history of suicide attempt and severity/hospitalization due to suicide attempt in SV adolescents compared to NSV adolescents. Likewise, the condition of survivor had a large effect size on history of suicide attempt and severity/hospitalization due to this fact, showing a significant incidence on these variables.

Likewise, the assessment of the impulse to self-harm and the level of suicidal risk identified that adolescents in the SV group were more likely to present high levels of these behaviors related to risk of suicide in contrast to adolescents in the NSV group.

## Figures and Tables

**Table 1 children-10-00549-t001:** Socio-demographic characteristics of adolescents as a function of suicide-survivor status.

Variable	Survivors (*n* = 79)	Non-Survivors (*n* = 369)	*X* ^2^	*p*	*f*
Age					
12 to 13 years	5 (6.3%)	54 (15%)	0.059	0.020	0.083
14 to 15 years	19 (24.1%)	84 (23.3%)			
16 to 17 years	34 (43%)	164 (45.4%)			
18 years	21 (26.6%)	59 (16.3%)			
Socioeconomic status					
Low	35 (44.3%)	120 (33.2%)	0.156	0.172	0.102
Medium	32 (40.5%)	184 (51%)			
High	12 (15.2%)	57 (15.8%)			
Sex					
Man	39 (49.4%)	184 (51%)	0.796	0.16	0.925
Woman	40 (50.6%)	177 (49%)			
Housing location					
Residential	55 (69.6%)	268 (74.2%)	0.400	0.401	0.116
Rural	24 (30.4%)	93 (25.8%)			

*X*^2^ (Chi squared); *p* (Student’s T); *f* (Fischer exact test).

**Table 2 children-10-00549-t002:** Adolescent risk factors as a function of survivor status.

Variable	Survivors (*n* = 79)	Non-Survivors (*n* = 369)	*X* ^2^	*p*	*f*	*d*	*r^2^*
Hospitalization due to attempt							
Yes	14 (17.7%)	11 (3%)	0.000	0.000	0.000	0.708	0.000
No	65 (82.3%)	350 (97%)					
Family history of attempt							
Yes	34 (43%)	47 (13%)	0.000	0.000	0.000	__	__
No	45 (57%)	314 (87%)					
Antecedent attempt							
Yes	19 (24.1%)	39 (10.8%)	0.002	0.002	0.000	0.432	0.000
No	60 (75.9%)	322 (89.2%)					

*X*^2^ (Chi-square); *p* (Student’s T); *f* (Fischer exact test); *d* (Cohen); *r*^2^ (coefficient of determination).

**Table 3 children-10-00549-t003:** Cognitive and affective empathy as a function of survivor status.

Variable	Dimensions	Survivors (*n* = 79)	Non-Survivors (*n* = 369)	*X* ^2^	*p*	*f*	*d*	*r* ^2^
Cognitive and affective empathy	Perspective adoption							
	Low	27 (34.2%)	170 (47.1%)	0.037	0.037	0.000	0.173	0.000
	High	52 (65.8%)	191 (52.9%)					
	Emotional understanding							
	Low	25 (31.6%)	134 (37.1%)	0.359	0.360	0.041	0.188	0.000
	High	54 (68.4%)	227 (62.9%)					
	Cognitive empathy							
	Low	37 (46.8%)	171 (47.4%)	0.932	0.932	0.856	0.111	0.000
	High	42 (53.2%)	190 (52.6%)					
	Affective empathy							
	Low	40 (50.6%)	149 (41.3%)	0.128	0.129	0.119	−0.064	0.000
	High	39 (49.4%)	212 (58.7%)					

*X*^2^ (Chi-square); *p* (Student’s T); *f* (Fischer exact test); *d* (Cohen); *r*^2^ (coefficient of determination).

**Table 4 children-10-00549-t004:** Impulse for self-injurious behavior and suicidal risk in adolescents as a function of survivor status.

Variable	Dimensions	Survivors (*n* = 79) *n* (%)	Non-Survivors (*n* = 369) *n* (%)	*X* ^2^	*p*	*f*	*d*	*r* ^2^
Self-injury impulse	Low	54 (68.4%)	303 (83.9%)	0.001	0.001	0.000	1.116	0.003
	High	25 (31.6%)	58 (16.1%)					
Suicide risk	Low	46 (58.2%)	277 (76.7%)	0.001	0.001	0.000	1.200	0.003
	High	33 (41.8%)	84 (23.3%)					

*X*^2^ (Chi-square); *p* (Student’s T); *f* (Fischer exact test); *d* (Cohen); *r*^2^ (coefficient of determination).

**Table 5 children-10-00549-t005:** Perceived social support of adolescents as a function of survivor status.

Variable	Dimensions	Survivors (*n* = 79)	Non-Survivors (*n* = 369)	*X* ^2^	*p*	*f*	*d*	*r* ^2^
Perceived social support	Family							
	Low	39 (49.4%)	173 (47.9%)	0.816	0.816	0.741	−0.020	0.000
	High	40 (50.6%)	188 (52.1%)					
	Friends							
	Low	45 (57%)	196 (54.3%)	0.666	0.667	0.321	−0.060	0.000
	High	34 (43%)	165 (45.7%)					
	Others							
	Low	36 (45.6%)	167 (46.3%)	0.911	0.911	0.814	0.000	0.000
	High	43 (54.4%)	194 (53.7%)					
	Global social support							
	Low	41 (51.9%)	175 (48.5%)	0.582	0.583	0.897	−0.080	0.000
	High	38 (48.1%)	186 (51.5%)					

*X*^2^ (Chi-square); *p* (Student’s T); *f* (Fischer exact test); *d* (Cohen); *r*^2^ (coefficient of determination).

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
