# Peer review of "Risk and Protective Factors in Ecuadorian Adolescent Survivors of Suicide"

_children, 2023, doi:10.3390/children10030549_

Round 1

Reviewer 1 Report

I read your manuscript entitled Risk and Protective Factors in Adolescent Survivors of Suicide with great interest. I think it is a well-written paper, worthy of investigation in a field where relatively little literature exists and of current interest.

The abstract explains the purpose of the paper directly and clearly.

The introduction and discussion are well articulated. The article thus presents a nice overview of the relevant literature on this topic and adds significant theoretical and practical implications, and I enjoyed reading it.

However, I have a few observations to report. 

With respect to the introduction I suggest that we could include some recent contributions on the topic of suicide ( see for example:  Chen, W., Boggero, A., Del Puente, G., Olcese, M., Prestia, D., Jahrami, H. & Bragazzi, N. L. (2021). Googling for Suicide–Content and Quality Analysis of Suicide-Related Websites: Thematic Analysis. JMIR formative research, 5(11), e2914 doi: 10.2196/29146).

I also think this statement needs to be revised as I believe it is incorrectly worded "The suicide phenomenon is so complex that no human being is exempt from having 36 some experience related to this issue throughout their lives, even in the United States it 37 has been identified that 51% of adults are exposed to at least one suicide during their life- 38 time, of which 35% have reported severe or moderate grief from the loss (Goulah-Pabst, 39 2021)."

A second observation concerns the procedure. In this section I think it is appropriate to include the approval of the ethics committee since psychology students are referred to as administrators. Also, it would be appropriate to include which institutions/groups were approached to then contact the participants. 

The analyses are carried out appropriately and the results are presented following scientific rigor and in a clear manner. Overall, it seems to me to be a good work, well structured and interesting. 

Author Response

Thank you very much for your comments, we are sure that this is a great contribution to our research. Here is a response to your observations:
1. In the results discussion section, a paragraph was included on the need for online presence of health professionals to contribute to the prevention of suicide in the adolescent population and I take into account the suggested bibliographic reference as can be seen in the reference number 45.
2. Revised statement about suicide exposure and corrected wording
3. In the procedure we include the code of the ethics committee that approved the execution of the research project and the procedure to contact the participants

Reviewer 2 Report

The manuscript addresses an interesting topic, focusing on an important mental health issue that affects young people. However, there are a few noteworthy problems:

1.       The title could inform that the study addressed Ecuadorian adolescents.

2.       The abstract must be improved.

·         The abstract must begin with the background "Background: Place the question addressed in a broad context and highlight the purpose of the study” (Children website).

·          When mentioning the two groups, it would be useful to describe the SV group.

3.       Introduction:

·         The introduction presents the main concepts and why the study is relevant. However, according to the Instructions for Authors of this journal, the specific hypotheses must be included.

·         Some paragraphs can be joined together, avoiding very short and single-sentence paragraphs.

4.       Method:

·         When describing the instruments, the authors must include the number of items of the Plutchik Suicide Risk Scale, as well as information about the psychometric properties of TECA.

·         I believe the software should be cited like this: IBM SPSS® 25.0 (IBM Corporation, Armonk, NY, USA).

5.       Discussion:

·         Include the reference when saying “However, other studies have generally focused on…”, as well as in the following sentence “In this line, other studies (xxxxxxxx)…” (line 232).

·         The limitations of this study must be addressed, as well as future studies must be suggested.

6.       The grammar of this manuscript must be revised.

Author Response

Thank you very much for your comments, we are sure this is a great contribution to our research. Below, we respond to his observations:

1. The title could inform that the study was aimed at Ecuadorian adolescents.
2. A paragraph with background of the problem is included in the abstract and from this a brief description of the SV group was made
3. In the introduction paragraphs 1 and 2 were joined; 7 and 8. In addition three hypotheses were included at the end of the introduction segment
4. In the method when describing the Plutchik suicide risk scale the number of items (15) was included, in addition the psychometric properties of the TECA are reported and the citation of the SPSS software was corrected
5. Missing references were included
6. Ser included a paragraph on limitations of the study (see lines 265-270)

Reviewer 3 Report

Dear authors, I have reviewed the submitted proposal. Although I consider it to be a relevant work that is in line with the special issue of the journal, I detect major limitations that, from my point of view, merit a complementary and preliminary study to the objectives described in the present report. 

The major and main problem is that, before being able to carry out comparative analyses between two different populations (SV vs. non-SV), it is necessary to show that the measurement models (factor structure) on which the measurements of the respective variables are based: a- fit adequately in both populations, and b- are invariant in both populations. Unfortunately, the small sample size of the SV group prevents these analyses from being carried out in a relevant way with the data that researchers currently have at hand. The correct approach would be to expand the SV sample to a number commensurate with the factor analyses needed to apply the factor and factor invariance analyses.  

On a minor note, I have detected words that have not been translated from Spanish to English, and acronyms that have been written according to Spanish and not English.

I wish you all the best in their future work.

Sincerely,

Author Response

Thank you very much for your comments, we are sure this is a great contribution to our research. Below, we respond to his observations:
1. Adolescent suicide survivors are a population with very particular characteristics that limit the possibilities access in developing research; on the one hand, they are exposed to the stigma due to which they tend to be invisible, and on the other, being minors their legal guardians are generally reluctant to be studied. For the case of this research all cases in a city were inquired from the hospital reports of the last 5 years and only adolescents whose legal guardians accessed participated. For the foregoing, extending the sample would involve waiting for a longer period of time for other suicides to present themselves and thus contacting the guardians of these minors.
2. In order not to miss the opportunity to offer findings about the study object population information the rater’s warning about the factor structure was included to be taken into account in future studies within the limitations of the study as can be seen in lines 265 a 269

Reviewer 4 Report

First, thank you very much for the opportunity to review an article that is extremely relevant to the field of adolescent health. The present study brought very interesting results and will only need some adjustments. Congratulations to all the authors for the excellent work.

1. Introduction

General coment: Please perform a careful punctuation review of the entire text of the article, not just the introduction. Also a review of writing in English.

2. Method

2.1 Describe further how these adolescents were recruited. In schools? If so, were all schools in the municipality or was there a draw? (line 90).

2.2 On line 103, about the instruments. I suggest authors check and make sure that all instruments used in the study were validated in Ecuador and then cite the source. I noticed that this procedure was performed only for one instrument.

3. Results

General comment: just to clarify, did the authors cut scale dimensions and present those results that gave statistical significance? Would they have additional data that was not presented? I would suggest that the authors bring more about the results of the scales, even if they are not statistically significant. The article was short, there is space to insert more information, since this topic is very relevant.

3.1 On line 175, check whether the English term "size" would be the most appropriate.

3.2 On line 191 the writing of the table title was confused, the terms "of", "and" together.

Author Response

Thank you very much for your comments, we are sure this is a great contribution to our research. Below, we respond to his observations:
1. In the introduction a general review of the punctuation marks and the English writing of the document was carried out
2. In the method a more detailed description of the procedure was carried out to recruit adolescent survivors and non-survivors
3. The assessment scales were retained in their factor structure as reported by their original authors
4. Reliability and validity information of all instruments is included
5. Additional information on the limitations of the study and suggestions for subsequent studies are included
6. The term “Effect size” is the indicated to explain the statistical procedure. More information can be found at: https://www.psychometrica.de/effect_size.html
7. The term “and” in line 207 was deleted

Round 2

Reviewer 3 Report

Dear authors, thank you for your response. When I sent my initial suggestion of rejection, it was precisely because the correct response to my suggestions merits collecting new data and, fundamentally, also rethinking the proposal considering the construct validity of the measurements used. This is also why I stand by my initial decision. I wish you all the best in your work.

Author Response

Thank you very much for your comments, we are sure that this is a great contribution to our research. Here is a response to your observations: 1. Survivors of adolescent suicide are a population with very particular characteristics that limit the possibilities of access in the development of research; On the one hand, they are exposed to stigma for which they tend to make themselves invisible, and on the other, as they are minors, their legal guardians are generally reluctant to allow them to be studied. In the case of this investigation, all the cases of a city were investigated from the hospital reports of the last 5 years and only adolescents whose legal guardians agreed to participate. Therefore, extending the sample would imply waiting a longer period of time for other suicides to occur and thus contact the guardians of these minors. 2. In order not to lose the opportunity to offer discovery about the population under study, information was included, the evaluator's warning about the factorial structure, so that it is taken into account in future studies within the limitations of the study, as can be seen in lines 265 to 269
